# Extracting Morphological and Sub-Resolution Features from Optical Coherence Tomography Images, a Review with Applications in Cancer Diagnosis

Christos Photiou [1,*], Michalis Kassinopoulos [2] and Costas Pitris [1]

[1] Department of Electrical and Computer Engineering, KIOS Research Center, University of Cyprus, Aglantzia Avenue 1, Nicosia 2109, Cyprus

[2] Institute of Neurology, University College London, London WC1N 3BG, UK

[*] Correspondence: photiou.christos@ucy.ac.cy

**Abstract:** Before they become invasive, early cancer cells exhibit specific and characteristic changes that are routinely used by a histopathologist for diagnosis. Currently, these early abnormalities are only detectable ex vivo by histopathology or, non-invasively and in vivo, by optical modalities that have not been clinically implemented due to their complexity and their limited penetration in tissues. Optical coherence tomography (OCT) is a noninvasive medical imaging technology with increasing clinical applications in areas such as ophthalmology, cardiology, gastroenterology, etc. In addition to imaging the tissue micro-structure, OCT can also provide additional information, describing the constituents and state of the cellular components of the tissue. Estimates of the nuclear size, sub-cellular morphological variations, dispersion and index of refraction can be extracted from the OCT images and can serve as diagnostically useful biomarkers. Moreover, the development of fully automated algorithms for tissue segmentation and feature extraction and the application of machine learning, can further enhance the clinical potential of OCT. When fully exploited, OCT has the potential to lead to accurate and sensitive, image-derived, biomarkers for disease diagnosis and treatment monitoring of cancer.

**Keywords:** optical coherence tomography; OCT applications; cancer; image processing; feature extraction; classification





## 1. Introduction

The majority of cancers originate in the epithelial layers of various organs throughout the body. Before they become invasive, at stages known as dysplasia and carcinoma in situ, early cancer cells alter the epithelial micro-structure as well as the cellular constituents. More specifically, the number of cells, and therefore the number of nuclei, increase and the epithelium becomes crowded. The nuclei become bigger, polymorphic, and hyperchromatic, features that are routinely evaluated during histological examination. Currently, these early abnormalities are only detectable by histopathology or, non-invasively, by optical imaging techniques such as confocal or multi-photon microscopy. Unfortunately, neither of these two techniques has been clinically implemented due to their technical complexity and their limited penetration in tissue. Optical coherence tomography (OCT) can address some of these limitations. OCT has been extensively used in medical imaging, especially for ophthalmologic pathologies, but also in cardiology, gastroenterology and tissue engineering. Further applications have also been developed in other fields, such as industrial non-destructive testing, optical fiber and polymer composite characterization and even in the jade industry [1–3]. OCT is also currently applied to the diagnosis of cancer in various tissues and organs [4–7].

*Optical Coherence Tomography (OCT)*

Optical coherence tomography (OCT) is a noninvasive medical imaging technique with increasing use in the diagnosis of disease. Images are formed by measuring light backscattered from the tissue microstructures. However, unlike ultrasound (US), OCT requires interferometric techniques, because of the high speed of light, to detect the origin of the reflected signal. The interferometer creates one-dimensional A-Scans that include the depth-dependent backscattered intensity in the sample along the beam's path. Two-dimensional, cross-sectional, OCT images of tissue are then constructed by juxtaposing a series of axial measurements at different transverse positions. The resulting data set is a two-dimensional array, which represents the optical backscattering within a cross-sectional slice of the tissue. Three-dimensional imaging can also be performed by stacking several two-dimensional cross-sectional images at different positions. In OCT, the axial and the transverse imaging resolutions are independent. The axial resolution is determined by the coherence length of the light source and is decoupled from the beam focusing which determines the transverse imaging resolution. Light sources appropriate for OCT exhibit a broad spectrum and, hence, a short temporal coherence length, high spatial coherence and can provide high-resolution imaging. Some examples include superluminescent diodes, femtosecond lasers and wavelength scanning sources [8].

OCT technology has significantly improved over the past few years with the introduction of fast, high resolution, OCT systems which are uniquely suited for in vivo applications. The first OCT systems to be developed were time-domain OCT (TD-OCT) systems and were first reported in 1991 for imaging of the human retina in vitro [9]. In TD-OCT the reference arm of a Michelson interferometer is scanned to determine the path length of the light backscattered from the sample. Interference occurs only if the sample path length equals the reference path length (Figure 1). For sufficiently short coherence length lasers, inhomogeneity regions within the sample will result in interference patterns. The major limitation of TD-OCT is that the imaging speed is restricted by of the need to scan the reference mirror. Since TD-OCT requires accurate mechanical scanners, the imaging speed cannot usually exceed one-hundred A-Scans per second.

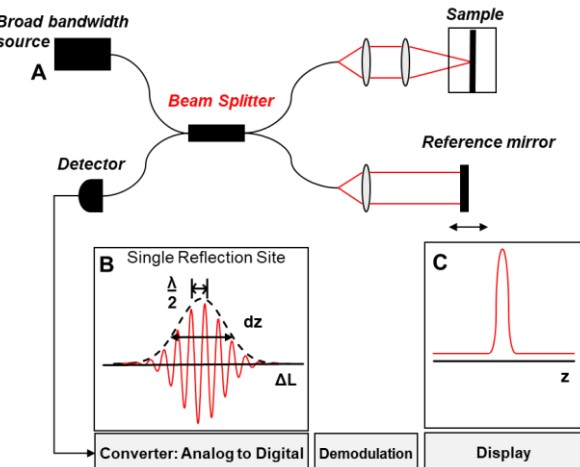

**Figure 1.** (**A**) Fiber-optic implementation of a time domain low coherence interferometer, (**B**) interferogram, and (**C**) the A-Scan envelope. Where ΔL is the arm's optical path length difference and dz the coherence length.

Fourier domain OCT (FD-OCT), proposed by Fercher in 1995, is based on Wolf's inverse scattering solution for objects with finite scattering properties [10]. In FD-OCT, there is no movement of the interferometer arm. The intrinsic morphology of the sample is encoded in the spectral response and is extracted by Fourier transformation (Figure 2). In addition, since all the backscattered light is detected simultaneously, the signal-to-noise ratio (SNR) and sensitivity of FD-OCT is superior to TD-OCT. Thus, FD-OCT allows much

higher scan speeds than TD-OCT systems [11]. A key FD-OCT limitation is that higher frequency features, in the resulting spectra, require a higher resolution spectrometer to support the same detectable range inside the sample, thus, imposing demanding requirements on the spectrograph and CCD camera [11]. The signals also display conjugate artifacts. As the signal of the spectrum is a real function, its Fourier transform is Hermitian, and the resulting image includes both the sample and its mirror image. These artifacts can be eliminated by phase shifting interferometry (PSI) also known as phase stepping interferometry [12,13].

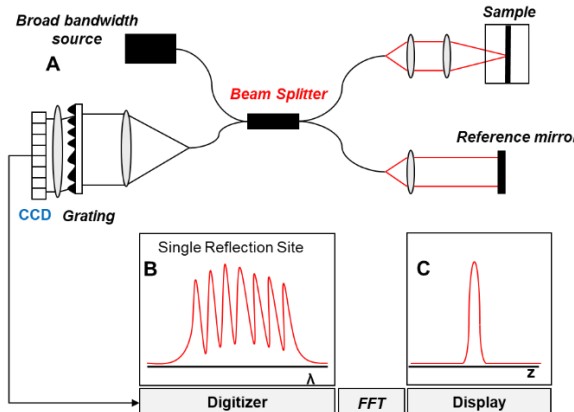

**Figure 2.** (**A**) Fiber-optic implementation of Fourier domain low coherence interferometer, (**B**) spectrogram and (**C**) back reflection profile.

Swept source OCT (SS-OCT) is a variation of FD-OCT, where the wavelength-dependent intensity data are recorded sequentially, with a single photodetector, while tuning the wavelength of the light source. As in the case of FD-OCT, the main advantage of this technique is that the reference arm length is fixed, and no moving parts are required. This significantly increases the speed of scanning. In addition, using two photodetectors, in a heterodyne configuration, provides the added advantage of easy rejection of the unwanted DC intensity terms and improved SNR (Figure 3). This enhances the usable dynamic range of the detection system considerably. The main disadvantage of SS-OCT is that the light sources are expensive and only available for a limited range of wavelengths. However, this limitation will probably be circumvented in the near future [14,15].

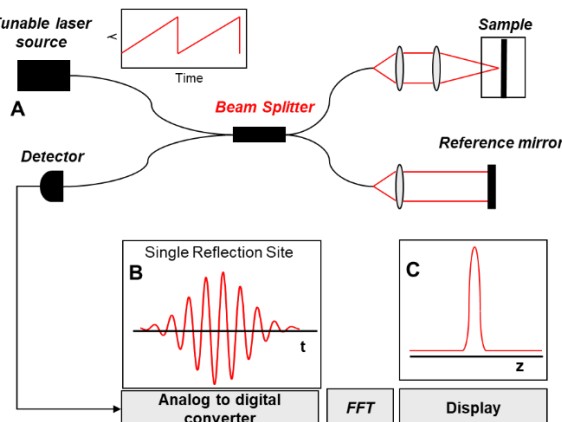

**Figure 3.** (**A**) Fiber-optic implementation of swept source low coherence interferometer, (**B**) interferogram, and (**C**) back reflection profile.

OCT imaging has several advantages that make it attractive for a broad range of applications. The axial (depth) resolution of OCT is usually ~2–15 μm, which is ~100 times

better than standard ultrasound imaging [16], OCT can, therefore, perform imaging at a resolution approaching that of histopathology. OCT can be implemented fiber-optically, using devices such as handheld probes, endoscopes, catheters, laparoscopes, and needles that enable non-invasive or minimally invasive in situ imaging. It can also perform imaging in real-time, making it suitable for guidance of excisional biopsies or interventional procedures. Finally, OCT is compact and portable, which are important features for a clinically viable device. The most developed OCT applications are those focusing on ophthalmology [7,17,18], cardiovascular pathologies [19,20], dermatology [4,21,22] and GI tract imaging [23–26]. OCT technology has also been utilized for surgery image-guidance in many medical disciplines [27–29]. In addition, OCT systems have been shown to be able to detect and diagnose different kinds of cancers [30–37]. However, some tissue and cellular changes, which are characteristic hallmarks of cancer, remain below the resolution limit of standard microstructural OCT imaging. Diagnosis is further limited by lack of contrast at the cell or tissue level. Fortunately, the clinical capabilities of OCT, leading to detection and diagnosis of malignancies, can be significantly enhanced by extracting additional features from the OCT images, which reflect changes in many intrinsic sub-resolution tissue parameters [15,38].

## 2. OCT Applications in Cancer Diagnosis

In cancer diagnosis, OCT could be utilized in three general scenarios. First, OCT could guide standard excisional biopsies to decrease false negative results. This could improve the accuracy of biopsy-guided diagnosis and reduce the number of biopsies required, resulting in better prognoses as well as significant cost savings. Second, it could be possible to use OCT to directly diagnose or grade disease. This application would be more challenging since it implies making a diagnosis based on OCT rather than relying on conventional histopathology. Applications could include scenarios where OCT would be utilized to grade early neoplastic abnormalities or detect the depth of neoplastic invasion. Third, diagnosis and treatment could be performed simultaneously, under real-time OCT guidance. This would require that the OCT diagnostic information be directly coupled to the treatment decisions. The integration of diagnosis and treatment could reduce the number of patient visits, resulting in an important decrease in health care expenses and enhanced patient compliance. Each of these scenarios demands an increasing level of OCT performance, not only of its ability to accurately image various tissues, but also, more importantly, to achieve the necessary level of sensitivity and specificity required for each clinical situation.

In dermatology, researchers have investigated the diagnostic accuracy of OCT in distinguishing basal cell carcinomas (BCC) in vivo. Those studies yielded promising results, with sensitivity and specificity ranging from 80 to 93 and 84 to 95%, respectively [39,40]. In addition, Ulrich et al., examined the diagnostic value of OCT combined with clinical and dermoscopic evaluation. Their study achieved improved diagnostic accuracy, compared with clinical and dermoscopic information alone, with sensitivity and specificity of 96 and 75%, respectively. However, even skilled observers found it very challenging to discriminate BCC from actinic keratosis, resulting in a 50% error rate [40,41]. Recently, researchers have utilized a new OCT approach, nanosensitive OCT, to demonstrate that differences between the nanoscale structure of normal and malignant regions, which are difficult to detect by conventional OCT systems, can be clearly distinguished [42–45].

Oral cancer, especially squamous cell carcinoma, is mainly treated by a mixture of surgery and radiotherapy. Because of the location and the neighboring structures of oral cancers, it is crucial to achieve complete tumor resection. OCT has been utilized in various studies to evaluate its potential to distinguish between malignant and benign oral tissue with promising diagnostic accuracies (82%) [46,47]. The morphological characteristics were verified by quantitative analysis, but no studies have yet been performed for real-time assessment of the surgical resection margins.

OCT has also been evaluated for the visualization of lung cancer during bronchoscopy and after surgery on resected specimens. Bronchial malignancies have been primarily defined on OCT images by a thickened epithelium wall and loss of sub-epithelial identifiable microstructures. Tumor invasion was indicated by a loss and/or disturbed architecture of the basement membrane [48–50]. While the diagnostic accuracy was quite high (81.8 to 83.3%), OCT during bronchoscopy is not yet sufficiently sensitive to completely replace biopsy. Although OCT has the potential to be integrated with bronchoscopic procedures for the diagnosis of lung tumors, effectiveness of tumor margin detection during surgery has not yet been assessed [51].

In the diagnosis and management of breast cancer, OCT has been assessed for tumor and sentinel lymph node classification. It has been shown to have a high diagnostic accuracy (84%) in boundary assessment, comparable to histology [52,53]. For lymph node detection, Full-field optical coherence tomography (FF-OCT) was capable of discriminating malignant invasion of lymph nodes from benign lymph nodes with high sensitivity and specificity (89 and 87%) [54,55].

Several studies have investigated the use of OCT for the diagnosis of ovarian cancer, two of which were undertaken during surgery. Tumor characteristics, based on qualitative image analysis, were extracted from SS-OCT or FF-OCT images [56,57]. Ovarian malignancies were distinguished by the appearance of hyperintense regions with irregular patterns, which turned out to be disorganized collagen fibers. Metastases could also be detected, as shown by Peters et al. [56]. In addition, some studies have focused on the optical coefficients, which were extracted from normalized histograms. After building a logistic classifier model, Nandy et al. were able to achieve 91.6% and 87.7% sensitivity and specificity for tumor classification from FF-OCT images [58,59].

OCT has also been applied effectively to the detection of gastrointestinal (GI) tract malignancies. Malignant and benign pancreatic duct strictures were discriminated both ex vivo and in vivo during basic endoscopic retrograde cholangiopancreatography (ERCP) procedures [60,61]. Biliary duct imaging with OCT was demonstrated by Arvanitakis et al., using specific criteria to detect malignant biliary strictures, with an accuracy of 84% in 37 patients [62]. OCT also showed promise in preoperative detection, compared with arbitrarily taken biopsies, with a 67% accuracy in the same cohort. Furthermore, Van Manen et al., assessed the accuracy of FF-OCT in distinguishing pancreatic tumors from surgical specimens and compiled specific criteria for different types of pancreatic tumors such as disruption of glands and the presence of tumor stroma [63]. Zhu et al. investigated the application of FF-OCT in hepatic specimens [64]. Regular hepatic structures, such as blood vessels and bile ducts could be very well distinguished and hepatic adenocarcinoma was identified by the presence of nuclear atypia and large tumor nodules separated by thick fibrous bands.

The challenges in the diagnosis of esophageal carcinomas, Barret's esophagus (BE), and dysplasia were also addressed using OCT. Bouma et al. performed the first in vivo study with 32 BE patients undergoing routine endoscopy and showed that BE is well distinguished from normal esophagus by visualization of disorganized glandular morphology and the absence of normal squamous mucosa [65]. In the detection of BE before and after radiofrequency ablation therapy, OCT was found to distinguish normal glands from buried Barrett's glands only in a small percentage of the patients (7.7%) (Figure 4) [66,67]. With ~80% sensitivity and ~65% specificity, OCT is presently not as accurate as histology for the detection of BE [68]. Dysplasia was identified in OCT images based on reduced scattering and tissue structure changes, which are currently the only available criteria [69–71]. Adenocarcinomas were also identified in OCT images with neoplastic epithelium. They contained large pockets of mucin surrounded by fibrotic and hypervascular tumor stroma. Occasionally, infiltration of heterogeneous structures into the muscular layers was mentioned as a feature of tumor invasion. Asymmetrical shape and crowding of submucosal glands were also indicative of the presence of adenocarcinoma (Figure 5) [68,72]. Identification of adenocarcinoma in patients undergoing upper GI endoscopy, with a recognition rate

of 95%, is indicative of the promise of OCT as a tool for the identification of esophageal malignancies [73].

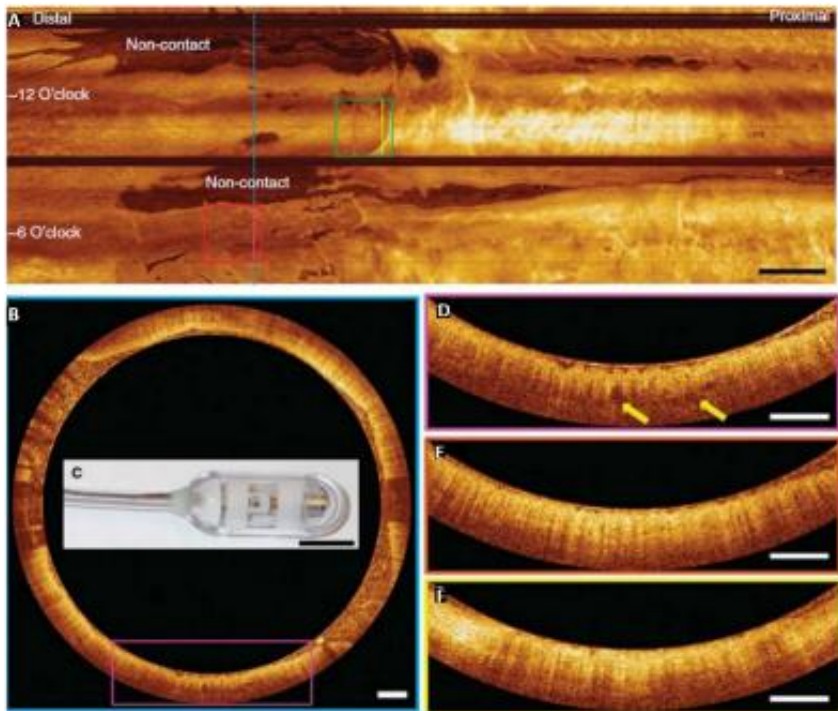

**Figure 4.** (**A**) *En face* OCT obtained below esophageal surface. Only the distal 12 cm out of 24 cm data are shown. (**B**) Representative cross-section (blue) from EMR region (red) (**C**). (**D**) Enlargement (pink) from (**B**) showing layer effacement, surface signal greater than subsurface, and multiple dilated glands (arrows). (**E**) Cross-section (brown) showing layered BE, which is likely non-dysplastic. (**F**) Cross-section (yellow) showing the squamo–columnar junction at a tongue of BE. Inset scale bars 1 mm. Reprinted with permission from [66]. Copyright 2016 Wolters Kluwer Health, Inc: American Journal of Gastroenterology.

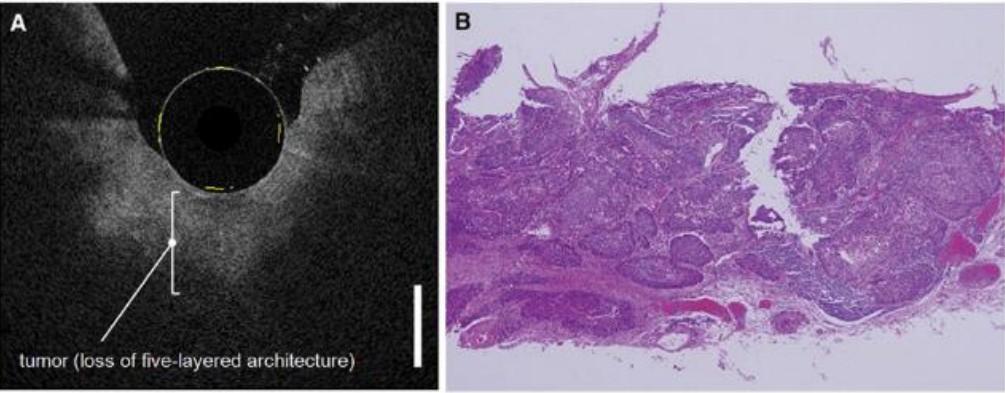

**Figure 5.** Example of endoscopic OCT of an esophageal squamous cell carcinoma. Corresponding OCT (**A**) and histology (**B**) image of tumor invasion in the submucosal layer, resulting in a loss of the five layered architecture (Bar = 1000 μm). Reprinted with permission from [74]. Copyright 2010 Elsevier: Gastrointestinal Endoscopy.

The application of OCT during colonoscopy has not been advanced as much as upper GI applications. Recently, quantitative analysis has been applied to both tumor and polyp detection, unfortunately, with low sensitivity. Tearney et al. [75] first demonstrated the application of OCT during colonoscopy for precancerous tissue identification. Imaging of

normal colonic wall features has also been performed by Westphal et al. [76]. For colorectal cancer, quantitative analysis has been performed in two studies, concluding that abnormal malignant tissue has a lower scattering coefficient and less variation of signal intensity from the surface, resulting in 78% sensitivity and 74% specificity [77,78].

Other studies have evaluated the diagnostic performance of OCT for the identification of cancer in bladder biopsies or surgical specimens in populations varying between 21 and 116 patients [79–82]. Distorted tissue layers and sub-epithelial nests of tumor cells were mainly found in biopsies of tumors. By using these criteria, sensitivity and specificity ranged between 84 and 100%, and 65 and 89%, respectively, for tumor recognition. Using FF-OCT, more details of tumor cells, such as the existence of large nuclei and newly formed blood vessels (as bright spots), could be seen. With appropriate training of the reviewers, disease diagnosis could be obtained with an accuracy up to 96% [82].

## 3. Intensity and Morphological Features for OCT Image Classification

Radiologists have been able to discriminate various tissues characteristics in OCT images using empirically derived image features. Some studies have even developed "radiological" criteria for interpreting OCT images. However, this approach has not been proven accurate enough to assure the high performance required to establish OCT as a commonly accepted clinical tool for cancer diagnosis. Therefore, there is an increasing effort to extract additional quantitative features from OCT images that can serve as biomarkers of disease. Furthermore, machine learning and deep learning have also been employed in an effort to find more accurate diagnostic approaches [47,48].

### 3.1. Texture Features

Textures are complicated intensity patterns or sub-patterns that have characteristic properties and can be seen as a similarity grouping in an image. Investigation of sub-pattern properties provides estimates of important characteristics such as lightness, uniformity, density, roughness, smoothness, etc., of the texture [83]. There are usually four main steps in texture analysis: (i) Feature extraction to estimate the various features of an image that describe its textural properties; (ii) Texture discrimination to divide the image into regions, each matching a particular homogeneous texture; (iii) Texture classification to specify if a segmented region belongs to a pre-defined set of classes and, finally, (iv) texture reconstruction to be able to rebuild the surface (3D) geometry from texture information.

Methods of texture analysis include structural, statistical, model-based and transform approaches. Structural approaches [83,84] use texture and a hierarchy of spatial arrangements of different primitives. To represent the texture, primitives and rules concerning placement must be appropriately defined and the probability that they are located in a particular region estimated accordingly. The resulting symbolic description of the image offers several advantages. However, this method is more applicable for synthesis rather than analysis. These complicated descriptions are not optimal for natural textures because of the variability between different tissue structures at the micro and macro scales and the lack of clear margins between regions. In medical image analysis, Serra et al., and Chen et al., proposed a tool, based on mathematical morphology, useful for feature extraction and classification [85,86].

Statistical approaches, as opposed to structural methods, do not attempt to explicitly understand the hierarchical structure of the texture. Instead, they identify the texture indirectly by the non-deterministic properties of the distributions and relationships between the image's grayscale intensity levels. Various techniques based on second order statistics (by pairs of pixels) have been reported to provide very good discrimination results [87]. Julesz et al., examined, for the first-time, the visual perception of texture in terms of the texture statistical characteristics [88]. They reported that the textures in gray scale images are distinguished effectively only if they vary in their second-order statistical moments. In contrast, third-order moments require much extra effort. Second-order statistical features for texture analysis can also be extracted from the co-occurrence matrix

proposed by Haralick in 1979. Various studies have confirmed that those features are very effective for texture discrimination in medical images. Also, multi-dimensional co-occurrence matrices have been shown to outperform wavelet packets when applied to texture classification [89,90].

Model-based texture analyses [91,92] try to define an image texture by generative image and stochastic models using fractal and stochastic approaches. The calculated parameters of the derived model are then utilized for image analysis. The most significant limitation of this approach is the computational complexity of the estimation of the stochastic model parameters. Fractal models have also been applied to natural texture modelling. However, they have weak orientation selectivity and are not applicable to describing local image morphology [90,93].

Transform techniques for texture analysis, such as Fourier, Gabor [94] and Wavelet transforms [95], transform the image to a space whose co-ordinate system is tightly connected to the properties of a texture (such as frequency or size). The two-dimensional discrete Fourier transform (DFT) can distinguish texture periodicity and orientation. The main drawback of techniques based on the Fourier transform is the lack of spatial localization resulting in poor performance in practice. Gabor filters provide better spatial localization, but their usefulness is diminished due to the fact that there is usually no single filter resolution at which one can localize a spatial structure in natural textures.

The wavelet transform has the advantage of representing textures at the most suitable scale by varying the spatial resolution. In addition, there is a wide range of wavelet functions, thus, the most appropriate can be chosen for texture analysis depending on the specific application. These advantages make the wavelet transform attractive for texture segmentation. However, it is not translation invariant, which is a significant limitation [96].

Texture analysis had not been applied to OCT images until the beginning of the 21st century. Now many studies argue for the importance of the textural features in the OCT images. Gossage et al., and Gao et al., used the spatial gray level dependence texture features (SGLDMs) and texture features derived from the two-dimensional discrete Fourier transform (DFT) for tissue classification of mouse OCT images and OCT images of human skin. SGLDM is the spatial histogram of an image that represents the distribution of gray scale levels for the calculation of the statistical textural features for the selected area. Features include: correlation, homogeneity, energy, entropy and contrast (inertia) [97,98]. Texture analysis of OCT images has provided very promising results, with sensitivities of 82% and 87%, and specificities of 69% and 74% for esophageal dysplasia detection and 92% sensitivity and 62% specificity in differentiating cancerous from noncancerous bladder tissue [69,99]. These studies were based on the hypothesis that the morphological structure loss of the normal histological organization, as a result of dysplastic tissue architecture transformation, is reflected in textural features, such as lightness, uniformity, density and roughness. Images with large homogenous areas, such as normal areas of esophageal tissue, also have large DFT feature values within the lower spatial rings. In contrast, images with small inhomogeneous regions, such as the crypt-like glandular structures in Barrett's esophagus, have large DFT feature values within the higher spatial frequency rings [97,100].

Another approach is the use of the center symmetric autocorrelation method (CSAC) with rotation invariant measures [70]. Furthermore, textural heterogeneity in OCT images, represented by the variance of intensity [101], has been utilized to identify various abnormalities, including gastric cancer [73,102,103]. Automated textural analysis has also been demonstrated for different ophthalmologic pathologies for image segmentation, quantification, and separation of layers for diagnostic purposes [104,105]. Similar approaches have been used to discriminate different types of skin dysplasia [106], ovarian cancer [59], esophageal malignancies [107] and coronary artery diseases [108].

### 3.2. Morphological Features

In contrast to texture features, morphological features emphasize different local characteristics of the image, such as the shape and borders. Marvdashti et al., in their study of basal cell carcinoma (BCC), extended this morphological analysis by incorporating the positional information associated with pixels within each image region in both the lateral and the axial directions. Analyses of both intensity and phase retardation data resulted in regions whose shape and spatial features extend in correlation with the histology-based classification [109]. Morphological features were calculated by segmenting each binary region of interest (ROI) into two through six regions.

In a study of gastrointestinal tissue disease, Garcia-Allende et al. applied morphological assessment of OCT images using intensity distributions of different image regions. For each image, regions were identified based on intensity [101]. Morphological feature analysis classification (MFAC) was also explored as a means to distinguish stomach malignant tissue [110]. The stomach images were examined only to a depth of 1.2 mm. Normal stomach tissue was a regular structure and every layer was homogeneous. Malignant tissue was heterogeneous with alternating high and low backscattering crypts that obscured the normal features. The contour lines at 100 depths within the depth of the tissue were used to extract quantitative features. Three such features were proposed for classification: the standard deviation at the 40th-pixel depth line, the standard deviation of the intensity line of $0.25 \times 20th + 0.5 \times 40th + 0.25 \times 60th$, and the standard deviation of all 100 contour depth lines. These morphological features adequately represented the layered structures and were markedly different in OCT images of normal and cancerous stomach.

Morphological features have also been applied to esophageal OCT images to distinguish different grades of dysplasia in Barrett's esophagus (BE) using endoscopic OCT (EOCT). A characteristic of dysplasia in images of Barrett's esophagus is reduced scattered intensity. For each A-Scan in the EOCT image, the backscattered power was a function of depth, and the optical characteristics of the tissue were used to identify malignancy. Qi et al., have reported that another visible characteristic of EOCT images, distinguishing dysplasia within BE, is the lack of a stripe-like pattern that can be seen in images of BE without dysplasia. Even though the reason these stripes appeared has not been determined, they have proved to be a very useful feature for tissue classification [69,70,110–112] (Figures 6 and 7).

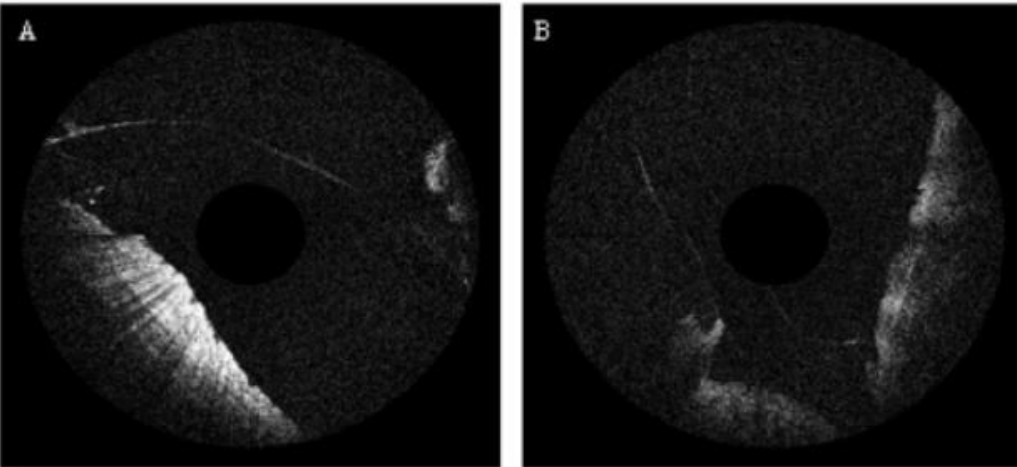

**Figure 6.** Stripe-like patterns in BE. (**A**) shows the obvious stripe pattern within a non-dysplastic BE endoscopic OCT image (EOCT); (**B**) shows no obvious stripe pattern within a high-grade dysplastic BE EOCT image [70].

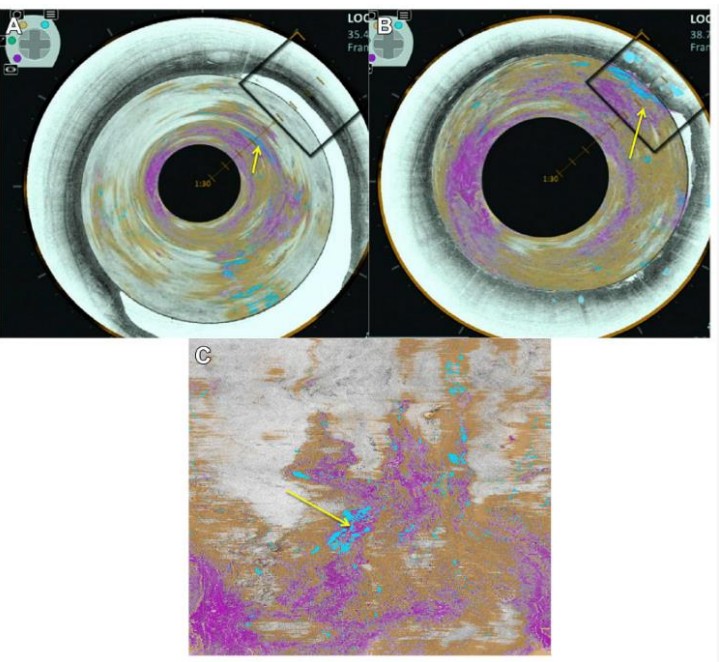

**Figure 7.** Esophagus image of the OCT showing a luminal en face view of an area of overlap (yellow arrow) between the three features of dysplasia (orange is lack of layering, blue is glandular structures and pink is a hyper-reflective surface). (**A**) A view looking down from the proximal esophagus. (**B**) A view closer to the suspected area of dysplasia. The en face view is also shown (**C**) [112].

### 3.3. Fractal Features

Mandelbrot proposed fractal geometry and the concept of the fractal dimension (FD) in 1967 [113]. An FD is the ratio providing a statistical index of complexity when comparing how details in a pattern change with the scale at which the pattern is measured. The box-counting method is the most common technique to calculate the FD in different fields because of its ease of use and because it is also adapted for OCT images. However, the box-counting method has been found to, occasionally, miscount the number of boxes, something that could lead to erroneous calculations of the FD [114,115].

The FD has been used in the analysis of OCT images to assess the structural variations of biological tissues. Fluearu et al. utilized the box-counting method to calculate the FD of porcine arterial tissue in an effort to distinguish and quantify variations in tissue texture at different locations in the OCT images [116]. Sullivan et al. used the same technique to calculate the FD for breast malignancy classification [117]. Furthermore, human skin studies have reported that melanomas had a larger FD than basal cell carcinomas and benign melanocytic nevi. The results could be explained by the fact that melanomas appeared to have heavily disorganized vessels with intense branching. Both the FD and the differential box-counting dimension could be utilized as an index to discriminate melanomas from the basal cell carcinomas and the benign melanocytic nevi [98,118,119].

In ophthalmology, the fractal properties of the retinal vasculature were evaluated for diagnostic purposes. Most studies used variations in the FD, characterizing the whole branching pattern of the retinal vascular network, to distinguish and diagnose eye disease [120,121]. Fractal analysis of OCT images has also been used to quantify photoreceptor rearrangement and vision restitution, recognize glaucomatous impairment at an early stage, and as an indicator of other pathological syndromes [122]. Furthermore, it has been applied to discriminate normal healthy eyes from diseased eyes with early neural loss in multiple sclerosis [123,124]. Somfai et al. [125] employed a power spectrum approach to perform fractal analysis of the layered retinal tissue to diagnose diabetic retinopathy. The study applied fractal analysis to each individual A-Scan of the segmented regions.

## 4. Sub-Resolution Features for OCT Image Classification

As mentioned before, various changes associated with early dysplasia are not resolvable in the intensity OCT images. That is a result of both the changes being below the resolution limit and the lack of optical contrast at the cellular and sub-cellular level. To remedy these issues, additional features can be extracted from the raw OCT signal, which correspond to sub-cellular and biochemical changes and could be used as disease biomarkers.

### 4.1. Group Velocity Dispersion (GVD)

Group velocity dispersion (GVD) is the physical phenomenon where different wavelengths of light travel at different velocities in the same medium due to the variation in the index of refraction as a function of wavelength. While an OCT interferometer can be optimized to compensate for dispersion from its optical elements, dispersion differences between sample and reference arms still emerge due to the variability and diversity of the properties of the tissues that are imaged. Interestingly, since dispersion is specific to the tissue that is causing the effect, it can convey useful information regarding its composition and/or constituent chemical concentrations. Given the dramatic changes in cellular biochemistry caused by cancer, which are discernible by other optical techniques such as Raman spectroscopy [126], it is highly likely that dispersion can also be used as a contrast mechanism in OCT imaging. Therefore, GVD could be used to detect, for example, changes associated with early cancer and result in more accurate disease diagnosis.

The concept of exploiting dispersion as a source of contrast is not new since there have already been examples where the dispersion of biomolecules is used to quantify their concentration. For example, the dispersion of hemoglobin has been used to extract a concentration of hemoglobin (Hb) from an intact red blood cell (RBC) [127]. This provided a simple and practical method, using spectroscopic phase imaging, to simultaneously measure Hb concentration and cell volume of living RBCs. In this case, dispersion provided molecular specificity with quantitative phase maps at different wavelengths that could differentiate among molecules. Furthermore, quantitative dispersion microscopy has confirmed that the dispersion of live HeLa cells agreed well with the dispersion measured for pure protein solutions [128]. Variations in the dispersion of different types of normal skin had also been identified in vivo using coherent reflection measurements of different skin types [129].

There are three approaches to estimate dispersion from OCT images: (i) measuring the resolution degradation [130,131], (ii) measuring the shift (walk-off) between images taken at different center wavelengths [132,133] and (iii) calculating the second derivative of the phase of the spectrum [134,135]. Each of these methods were evaluated, ex vivo, with images from samples with different scattering properties as well as tissue images [136,137]. However, all these approaches require that a strong, distinct, reflector is present in the image which is rarely the case in living tissue.

A new technique for estimating the dispersion using the speckle in OCT images to calculate the resolution degradation, which, thus, does not rely on distinct and strong reflections, has recently been proposed. Since speckle is present in most biological samples, this technique is applicable to any tissue and can be implemented in vivo and in situ. This technique is based on the fact that a portion of an OCT image that contains speckle, close to the surface of the sample, is related to a similar portion of the OCT image at a given depth by a depth-dependent speckle-degrading impulse response. By applying a Wiener deconvolution approach, the speckle degrading impulse response can be extracted from those image regions, leading to an estimate of the resolution degradation as a function of depth which can then be used to estimate the GVD [137,138]. The proposed method was verified ex vivo with GVD on various materials and tissues. Furthermore, its applicability to cancer diagnosis was evaluated on a small set of gastrointestinal (GI) normal and adenocarcinoma OCT images resulting in 93% sensitivity, 100% specificity and 96% accuracy (Figure 8) [136].

In another novel approach, tissue dispersion was estimated by calculating the cross-correlation of images acquired at different center wavelengths which provided an estimation of the dispersion-dependent wavelength shift. This shift, also known as walk-off, was then used to calculate the dispersion. Since a distinct reflector is not required for this method either, this approach is also applicable to any human tissue in vivo and in situ. This technique is based on the fact that walk-off will be present in two OCT images reconstructed from each of two halves of the spectrum. The cross-correlation of corresponding A-Scans in the two half-spectrum images provided an estimate of the walk-off shift, using the distance of the first peak to the zero-lag location. The GVD was, then, calculated using the walk-off [139]. The proposed technique was verified ex vivo resulting in GVD values comparable to those obtained from estimating the walk-off using a mirror (Figure 9). Furthermore, the method's applicability to cancer diagnosis was evaluated on a small set of gastrointestinal normal and cancer OCT images. Using the statistics of the GVD estimates, tissue classification resulted in 100% sensitivity, 81% specificity and 92% accuracy (Figure 10) [137,139].

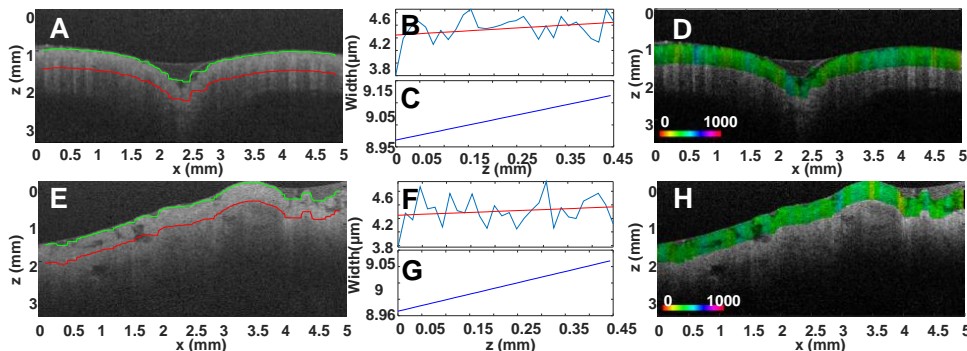

**Figure 8.** (**A**) OCT image of normal colon tissue (green line: top surface, red line: 0.5 mm depth). (**B**) Mean width as a function of depth for (**A**). (**C**) Degraded Gaussian width as a function of depth calculated from (**B**). (**D**) Overlay of the OCT image (gray scale) and the GVD for each A-Scan in a pseudo-color hue scale. (**E**–**H**) The same as before for colon adenocarcinoma [136].

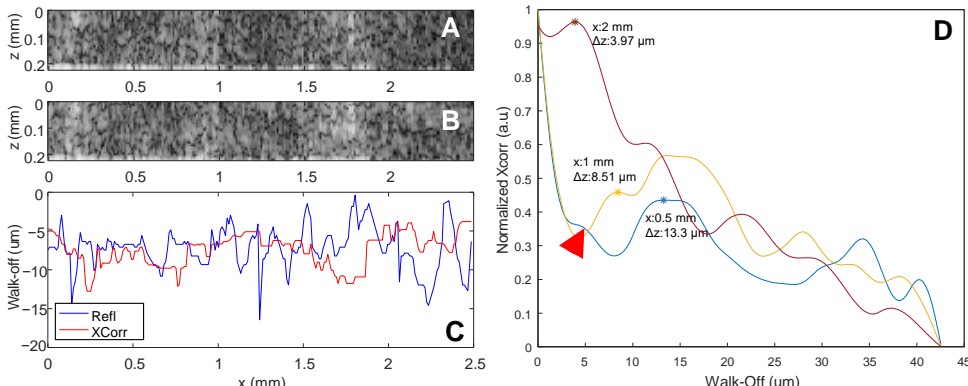

**Figure 9.** (**A**) Portion of the first half-spectrum OCT image from just above the bottom surface of the sample. (**B**) Similar portion from the second half-spectrum OCT image. (**C**) The walk-off for the 250 A-Scans in (**A**) and (**B**) calculated from the cross-correlation (red line). The blue line is the reference walk-off from the reflector. (**D**) Three indicative cross-correlation curves from different locations (x): 0.5 mm (blue), 1 mm (yellow) and 2 mm (red). The stars indicate the first maximum and the associated walk-off ($\Delta z$). The red arrow points to the location where the maximum should occur, which was missed due to weak cross correlation, a cause of error in the estimations [139].

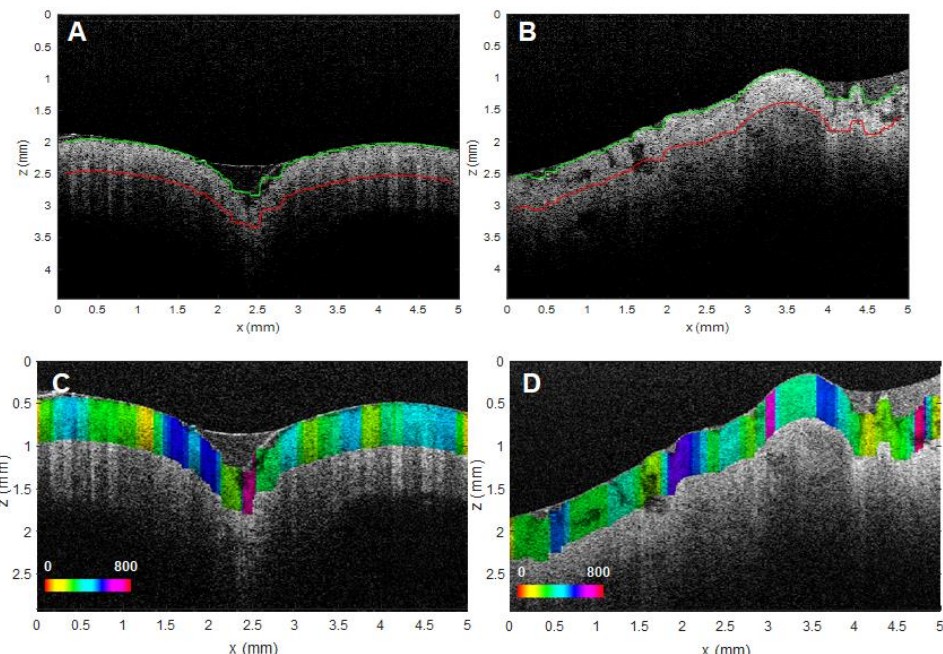

**Figure 10.** Normal (**A**) and abnormal (**B**) OCT images of human colon with the portion of the tissue used (green and red lines). Overlay of the images and GVD (pseudocolor hue, 0–800 fs$^2$/mm) for each A-Scan normal (**C**) and adenocarcinoma (**D**) [137,139].

### 4.2. Index of Refraction (n)

The index of refraction (*n*) depends on the speed of light in a medium and can be calculated by the division of the speed of light in vacuum, *c*, by the speed of light, *v*, in the medium. The use of *n* as an important intrinsic biomarker has already been recognized. Classic phase contrast or differential interference microscopy use *n* values as an optical imaging contrast. However, they do not provide a detailed mapping of *n* distributions in tissue. Rather, they are limited to thin slices of samples that are fixed on slides or cell cultures. Recently, there has been an increasing interest in measuring 3D *n* distributions in various applications in bioimaging, since *n*, as an intrinsic optical parameter, could provide label-free cell imaging with quantitative information about the sample.

Recent studies have shown that *n* could be used as a biomarker for medical diagnosis. Microbiology studies have shown that *n* distribution reflected cellular growth and division and could also be used for bacteria identification [140,141]. Past studies have also reported that 3D *n* tomograms of individual bacteria could be measured using holotomography (HT), a technology that directly provides measurements of the 3D *n* distribution of a cell. Using a Mach–Zehnder interferometer and illumination scanning, 3D *n* tomograms of bacteria, extracted from a sample of *E. coli*, have been demonstrated [142]. Recently, white-light diffraction tomography has also been utilized to image 3D *n* tomogram of *E. coli* [143] and a magnetostatic bacterium which produces magnetic particles (magnetosome) [144]. In hematology, different types of cells have been identified by their *n* value. Bloodborne infectious diseases (e.g., malaria) or chronic human disease (e.g., sickle-cell anemia, diabetes mellitus) could be identified by their *n* distributions [145–147]. Lipid droplets (LDs) in hepatocytes have been implicated in various pathologies such as cancer and diabetes mellitus [148]. LDs could be effectively visualized by exploiting their *n* value. The index of lipid is significantly higher than protein and thus LDs could be identified by performing 3D *n* tomograms [149]. Measurements in cancer specimens have shown that the presence of cancerous cells modified the *n* distribution in breast, prostate and epithelial cancer [150–152]. Recently, label-free tissue imaging, utilizing *n* variations, was also adapted to neuroscience. Index of refraction distributions could be helpful in the

diagnosis of neurological diseases such as Alzheimer's [153] and Parkinson's diseases [154]. However, there are some limitations when using *n* distributions for medical diagnosis. First, it is difficult to relate *n* to molecular information, mainly because proteins have similar *n* values regardless of their type. Another challenge is that the distribution of *n* values can provide limited morphological information about tissue structure. Additionally, cells have refractive indices similar to their environment and this can result in a significant lack of contrast.

Several methods have been developed to estimate the *n* and actual sample thicknesses from the optical pathlength measurements of low coherence interferometry (LCI) or OCT [155]. Tearney et al., in 1995, proposed an in vitro method to determine the *n* of a sample by placing it on top of a planar reflecting surface and acquiring an OCT image of both. The difference between the pathlength in air and through the tissue could be used to estimate *n* [156]. The main limitation of this technique is that it can be applied only to excised samples.

Focus tracking has also been used for *n* measurements. Ohmi et al., in 1997, performed measurements on z-cut sapphire and glass plates. By moving the object and scanning through the sample they estimated an *n* accuracy of ≤0.3% [157]. Applications of the same technique on biological samples of animal tissue, and of human tooth and nail resulted in *n* measurements ranging from 1.37 to 1.65 and an accuracy of ±1% [158]. However, *n* was assumed to be homogeneous through the sample thickness. Confocal scanning has also been evaluated in combination with FD-OCT to estimate *n*. Optical thicknesses were estimated from the Fourier transform of the spectral data. Both the index of refraction and thickness could be calculated by moving the objective lens to focus on the top and bottom surface of a single layer object. This experimental procedure has been verified using water, air and oil solutions, inside a 250-μm deep container, and the results fell within a 0–1% error [159]. In the early 2000′s, Maruyama et al. proposed a new formulation of the chromatic dispersion, in terms of the phase index, and showed that both the phase and group refractive index and physical thickness of a single layer sample could be estimated without using a special sample holder [158]. The applicability of the focus scanning method to multi-layer samples has also been demonstrated using a sample made of 13 layers of glass cover slips and air [160]. However, this method is also not applicable to complex samples such as tissue.

The *n* can also be estimated by inserting the sample into one of the arms of a Michelson interferometer. Fochs, in 1950 [161], reported using a white-light Michelson interferometer and a spectrometer to record the spectral response pattern to measure *n*. More recently, both the thickness and group *n* were estimated using OCT and/or LCI [162]. Most studies have concentrated on measuring the *n* of materials, e.g., poly-methyl-methacrylate glass (PMMA) [163], fused silica [164], or other single-layer objects [165]. The main limitation of this approach is that the sample must be thin and transparent.

The methodologies described so far are not appropriate for in vivo imaging since they require either a mirror below the sample, a thin and/or transparent sample, or an otherwise complicated imaging setup and algorithm. A new measurement technique, which uses two OCT images obtained at different incidence angles and could be applied for in vivo estimation of *n* has been recently proposed. This approach is based on the fact that the path length observed in a sample is different in two images obtained at different incidence angles and directly depends on *n*. Therefore, the *n* can be estimated by measuring the path length changes and the incidence angles. To accurately and robustly estimate the path length difference, the cross-correlation of corresponding A-Scans in the two images can be used, as in the calculation of the walk-off due to dispersion. This dual-angle method has been validated experimentally using both clear and scattering samples (Figure 11). The resulting measurements of *n* were within a mean of ~1% of the expected values [166].

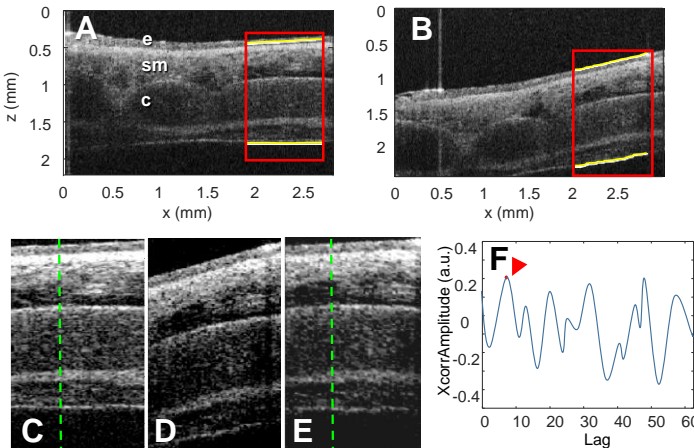

**Figure 11.** (**A**,**B**) The original OCT images of freshly excised trachea, at different incidence angles. The labels refer to the epithelium (e), the submucosa (sm) and cartilage (c). (**C**) The portion of image (**A**). (**D**) Portion of image (**B**). (**E**) The registered version of (**D**). (**F**) The cross-correlation of the A-scans indicated by the dashed lines in (**C**,**E**). The arrow points to the first maximum after the zero lag (not shown here for better visualization of the peaks) [166].

*4.3. Scatterer Size (SS)*

Mie theory predicts that scattering from particles with sizes comparable to the wavelength of illumination will exhibit specific spectral variations that are directly related to scatterer size. Based on this principle, there have been various attempts to estimate the size of epithelial cell nuclei using light scattering spectroscopy (LSS) [167,168]. Recently, LSS has been combined with low-coherence interferometry (LCI), providing depth localization of the LSS signal [169,170]. OCT has also been proposed for spectroscopic, depth-resolved, imaging. Spectroscopic OCT (SOCT) studies the localized spectra that are inherently available in the OCT signal [171], providing an additional contrast-enhancing mechanism. More recently, spectroscopic metrics that could precisely measure the size of cell nuclei, the main scatterer in epithelial tissues, have begun to be investigated.

Most of the approaches reported so far are based on the assumption that epithelial cell nuclei are spheroidal scatterers whose interactions with light are described by Mie theory [149]. Therefore, one approach to obtaining their nuclear size is to curve-fit the backscattered spectra to the Mie theory curves [172]. The main limitation of this approach is that it requires an exhaustive search through many possible scattering sizes and precise knowledge of the index of refraction of the scatterer and the ambient medium [173]. Furthermore, it does not sufficiently account for the spectral changes that could appear in experimental measurement in which the incident beam waist spot size is small or similar to the wavelength size. In addition, Mie theory was not formulated for Gaussian beams but rather for plane waves [174].

According to Mie theory, the "oscillation frequency" of the spectrum increases with increasing particle size. In the Fourier transform of the OCT spectrum the position of the maximum is indicative of the dominant scattering features in the region under observation [170]. In various studies of phantoms and biological tissues, this approach has resulted in accurate calculations of the scatterer sizes [170,173,175]. However, when the spectrum of the light source is narrow, the number of oscillations in the backscattered spectrum is not sufficient to discern the maximum peak among all the other, low-frequency, components [176]. Another approach for the measurement of the scatterer size, reported by Adler et al., is based on estimating the autocorrelation width of the backscattered spectrum [177]. This relies on the observation that backscattered spectra with high spectral modulation result in autocorrelation functions that shift quickly away from the central point while the autocorrelation functions of spectra with low spectral modulation are broader. Utilizing the width of the autocorrelation central lobe, the contrast of the OCT

images was enhanced. The main advantage of this method is that it is not very sensitive to sources of spectroscopic noise. Still, it does not always result in a precise estimation of the scatterer size. Other variations of the above methods have also been implemented. Oldenburg et al., utilized the autocorrelation width of the backscattered spectra, at 80% of the peak value, in order to improve the contrast of OCT images of macrophages and fibroblasts [176], while Kartakoullis et al., and Jaedicke et al., used the spectral information with principal component analysis (PCA) and clustering algorithms in an effort to discriminate phantom samples of microspheres with various diameters [178,179]. Tay et al. proposed that the use of many distinct spectral bands could enhance the sensitivity of scatterer size estimates, applying the method to spectroscopic OCT images of solutions of 0.5 and 45 μm microspheres which could be clearly discriminated [180].

Kassinopoulos et al., used Mie theory to create a new metric for SOCT, the correlation of the derivative (COD) bandwidth [181]. The feasibility, accuracy and robustness of this new method, in estimating scatterer size, has been confirmed utilizing images of microsphere phantoms and normal and cancerous human gastrointestinal tissue. The concept behind this approach is presented in (Figure 12) where the normalized backscattered spectra for scatterers with diameters of 6.0, 10.0 and 16.0 μm in the wavelength range of a SS-OCT system are shown. The characteristic oscillations are easily distinguishable in the spectra. The COD bandwidth method estimates the scatterer size by calculating the first derivative of the spectrum followed by the calculation of its autocorrelation. Subsequently, the lag position of its first minimum, referred to as the bandwidth of the correlation of the derivative (COD) bandwidth, is used as a metric for scatterer size estimation using a relationship derived from Mie theory. This method has also been used for the discrimination of normal vs. malignant esophagus tissue [182,183].

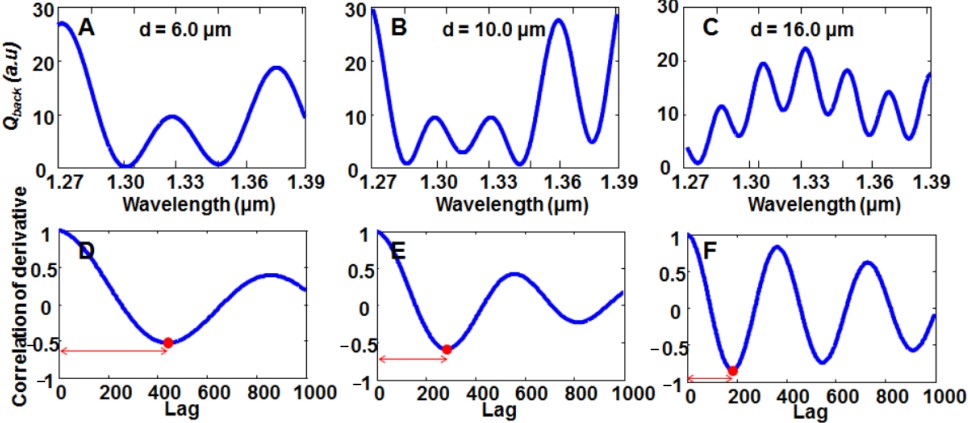

**Figure 12.** Backscattering Mie spectra for (**A**) 6 μm, (**B**) 10 μm and (**C**) 16 μm scatterers with medium and sphere refractive indices set at 1.47 and 1.59 respectively. The parameters for the calculations were chosen according to the specifications of the light source and the microsphere samples used in the experiments. Graphs (**D**–**F**) show the correlation of the derivative (COD) with the red dot indicating the first minimum and the red arrow indicating the bandwidth of the COD [181].

### 4.4. Nanoscale OCT

The majority of basic pathogenic processes in living tissues can cause alterations at the nanoscale. Recently, optical imaging technologies have been developed for imaging below the resolution limit [44]. However, these techniques are only capable of superficial 2D imaging, heavily rely on labelling, and are not appropriate for in vivo applications. The capability of OCT to detect nanoscale structural changes in weakly scattering media is restricted to the microscale. This limitation could be addressed by nanosensitive OCT (nsOCT), introduced by Alexandrov et al. in 2014 [42]. The nsOCT technology delivers nanoscale sensitivity to the structural level. However, it changes from one frame and estimates the sub-wavelength structure without resolving it [184,185]. nsOCT has already

been used for real-time and super-resolution contrast enhancement [186]. The technique is based on the fact that each spatial frequency component of the scattering object is represented at a specific wavelength of the optical illumination. Structural characterization of each object volume of interest was achieved with nanoscale sensitivity by determining the peak energy contribution in the spectrum which encoded the corresponding axial spatial period. Experimental results demonstrated that the nsOCT can distinguish changes within 3D objects, detecting nanostructural changes throughout a range of spatial periods. Dey et al. used a broadband supercontinuum laser with an nsOCT device to demonstrate fine sensitivity to nanoscale structural changes in ex vivo human skin tissue. Their results demonstrate that normal and malignant regions, which are difficult to delineate by typical OCT systems, could be clearly distinguished with nsOCT (Figure 13) [43,44].

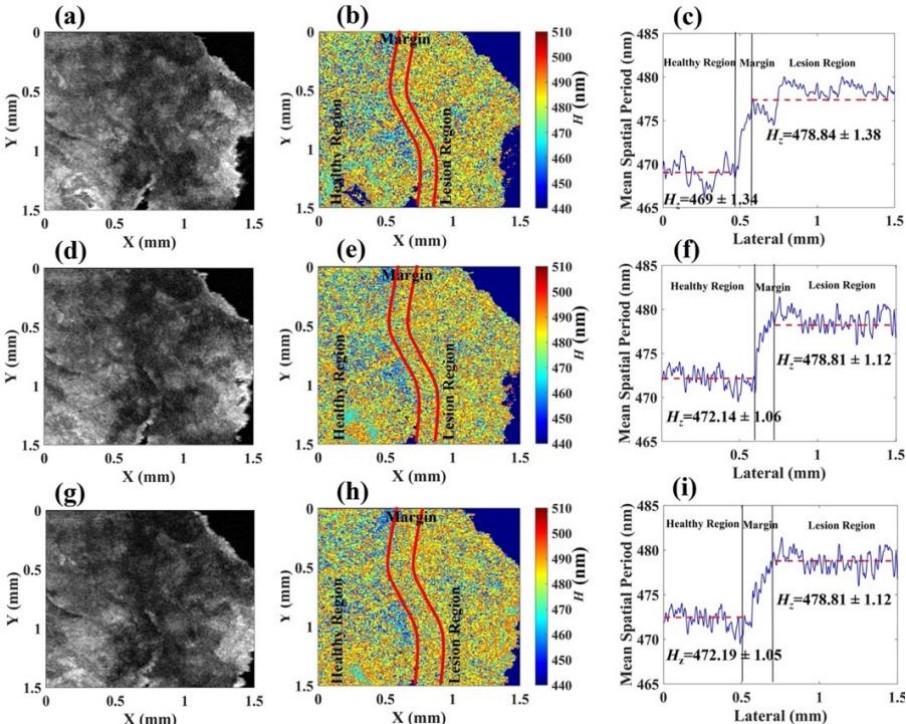

**Figure 13.** (**a**,**d**,**g**) Intensity-based OCT images of human skin at the depths of ∼200 μm, 400 μm, and 600 μm respectively. (**b**,**e**,**h**) The corresponding nsOCT en face images with spatial period mapping, presenting the dominant structural size. The healthy, marginal, and lesional regions are separated by the red lines. The color bar of the nsOCT en face images represents the spatial periods in nanometers. Plots (**c**,**f**,**i**) show changes in the mean dominant spatial period of the structure between the two regions, defining an intervening marginal area in the nsOCT en face images [44].

## 5. Segmentation and Classification of OCT Images for Cancer Diagnosis

Clinicians evaluating OCT images of different tissues, usually try to perform disease diagnoses manually, expending valuable effort and time [29,187]. Image assessment for morphological changes is the most common processing method and is routinely utilized for OCT image analysis. However, the performance of morphological analysis methods is restricted to simple and low-noise images. For larger images with complicated patterns, these operations require long computational times while, at the same time, their performance is further limited, due to the image variations [188]. OCT image assessment using machine learning (ML) has been, recently, introduced, especially in volumetric data to address various challenges pertaining to classification, discrimination and segmentation. ML can be performed after feature extraction, e.g., shape models such as statistical shapes [189] and deformable models [190], or pixel/portion level features. The latter can reach encouraging

performances, while at the same time, be less influenced by image characteristics such as noise and artifacts of segmentation tasks [191].

Qi et al., developed computer-aided diagnosis algorithms for the identification of dysplasia in Barret's esophagus (BE) using an endoscopic OCT system (EOCT). They evaluated the performance of four types of multivariate analysis: linear and quadratic discriminant analysis (LDA and QLA), K-nearest neighbor (*k*-NN), two types of neural networks (NNs) and classification trees. They achieved an accuracy of 84% for the classification of non-dysplastic vs. dysplastic BE tissue [69]. Another automatic segmentation and characterization method, applied to the esophageal wall in in vivo OCT images, resulted in an A-scan line classification accuracy of 94% with a sensitivity and specificity of 94% and 93%, respectively (Figure 14) [192]. Use of the discrete wavelet transform to analyze and characterize the local spatial distribution of gray levels in the images of BE was also investigated. Features extracted from the wavelet components found to be statistically significant, based on a two-sample t-test, were used as inputs to a Naïve Bayes classifier. Leave-one-out cross-validation, applied using an image volume of 60 images from 38 patients for BE vs. dysplasia classification, resulted in 86% sensitivity and 93% specificity [193]. Swager et al. used 60 images (from 29 endoscopic resections) from BE patients to detect early neoplasia in BE, comparing different ML classifiers and using leave-one-out cross-validation. Three new features were proposed: signal decay statistics, layering, and signal intensity distribution. The first exhibited the optimal performance with an AUC, sensitivity and specificity of 91%, 90% and 93% respectively [194]. Different machine learning methods for classification of esophageal tissue were also compared in the context of an increased range of features, which included intensity-based statistics, group velocity dispersion (GVD) and the average scatterer size (SS) of each A-scan. The comparison showed that a neural network-based approach, using leave-one-out cross-validation, provided the best performance, separating BE from dysplasia, for individual A-scans, with an accuracy of 89% [182] (Figure 15).

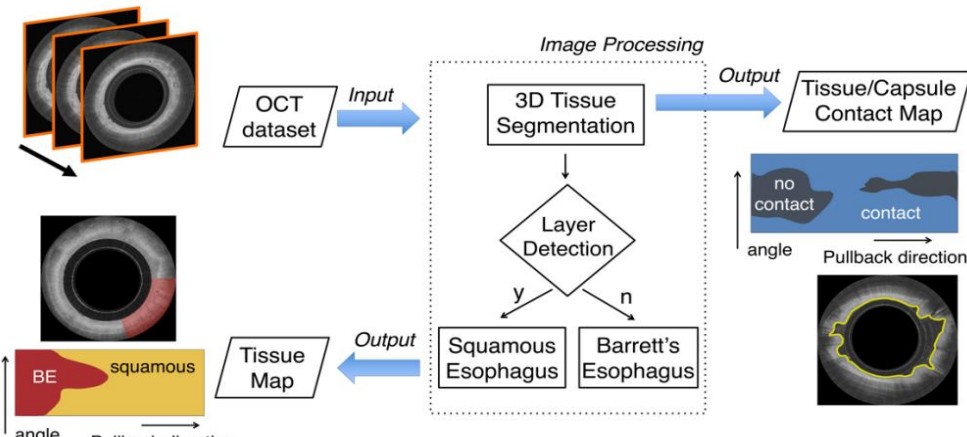

**Figure 14.** Flowchart of the entire automated processing framework. The algorithm receives as its input an entire OCT pullback data set that typically comprises > 1000 images. Initially, the algorithm automatically locates the position of the tissue's surface over the entire 3D data set and then tissue characteristics are assigned. Squamous esophagus is differentiated from BE (with or without dysplasia) tissue by identifying the presence of horizontal layers. The output of the algorithm is a tissue map of the entire data set, automatically depicting the presence of BE and a contact map showing areas that lack contact between the capsule and the tissue. [192].

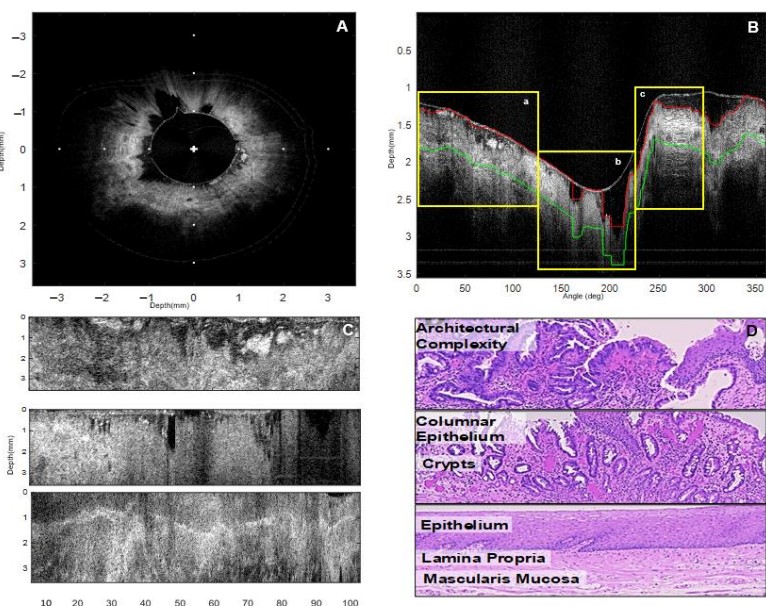

**Figure 15.** (**A**) In vivo OCT image of the human esophagus in Cartesian coordinates. (**B**) Same image in polar coordinates with the red and green lines indicating the top and bottom borders of the automatically segmented epithelial region (segmentation depth ~0.55 mm). The yellow boxes indicate annotated dysplastic (a), BE (b) and normal (c) regions. (**C**) Zoomed regions corresponding (from top to bottom) to the dysplastic, BE and normal annotated areas respectively. (**D**) Histopathologic sections (from unrelated samples) that illustrate the microstructural and nuclear changes associated with (from top to bottom) dysplastic, BE and normal esophageal tissue [182].

Morphological feature analysis (MFAC), in combination with ML, to identify stomach cancer was also investigated. First, five quantitative morphological features were extracted from OCT images. Then, five ML classifiers were evaluated for the classification (SVM's, the *K*-nearest neighbor, random forest, logic regression, and conventional threshold method). The results showed that features specifically created for stomach malignancies were significantly more effective than other commonly used morphological image features with over 95% accuracy for all five classifiers [110]. In addition, a deep learning method to automatically classify normal vs. cancerous gastric tissue in OCT images achieved an accuracy of 99.90% [195]. Allende et al., using a small number of samples, applied morphological analysis of OCT images for automated classification of GI tissues using several ML approaches with sensitivity values up to 99.97%, specificity up to 99.85% and accuracy up to 99.88% depending on the features selected for classification [101].

Recently, deep learning (DL) approaches have begun to be increasingly applied to OCT images of the gastrointestinal tract. Fonolla et al. trained an ensemble of deep convolutional neural networks to detect neoplasia in 45 BE patients. Their results show 95% sensitivity, 85% specificity, accuracy of 88% and an AUC of 0.95 [196]. A fully automatic multi-step computer-aided detection (CAD) algorithm, by Van der Putten et al., optimally leveraged the effectiveness of DL strategies by encoding the principal dimension in esophageal data. They trained and tested their algorithm using data from 23 patients to detect neoplastic regions in BE. Using the encoded principal dimension, they obtained an AUC of 0.93 with sensitivity and specificity of 85% and 92% respectively [197]. Unfortunately, these studies have two very important limitations. First, they use per image or per A-scan rather than per patient cross-validation that severely biases the results towards increased accuracy. It is important to perform the classification with leave-one-out cross-validation to avoid bias when the classifiers with A-scans or images from the same patient are trained and tested, especially when on a small dataset. Furthermore, no study performed normal vs.

abnormal tissue classification, which is very important for screening procedures where image acquisition is performed without endoscopic guidance.

A critical limitation of DL is the need for a large number of images which, many times, are not available. To alleviate this problem, data augmentation is applied. Gan et al., proposed an adversarial learned variational autoencoder (AL-VAE) to generate high-quality esophageal OCT samples. Their approach combined a generative adversarial network (GAN) and a variational autoencoder (VAE). Preliminary results show that the proposed method achieved better image quality in generating esophageal OCT images, when compared with the state-of-the-art image synthesis and discrimination performance networks. Using similar techniques, Wang et al., introduced an adversarial convolutional network, which uses adversarial learning to train a convolutional network for esophageal OCT image segmentation. The proposed framework takes advantage of pixel classification and adversarial learning, thus generating human-like segmentation results [198,199]. H. Luo et al., constructed a miniaturized OCT catheter and combined it with a residual neural network (ResNet)-based DL model to perform automatic image processing and real-time classification of normal and cancerous colorectal OCT images with 0.975 AUC (Figure 16) [200].

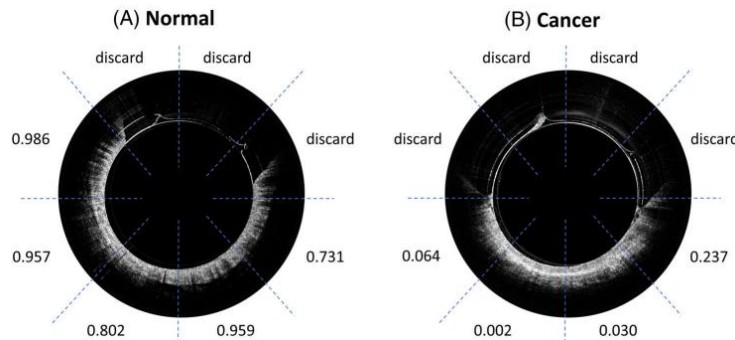

**Figure 16.** (**A**) ResNet prediction of OCT endoscope image of the normal region and (**B**) OCT endoscope image of the cancerous region. Reprinted with permission from [200]. Copyright 2022 John Wiley and Sons: Journal of Biophotonics.

ML and DL have also been applied to OCT images of various other tissues. Lenz et al., have demonstrated the application of *k*-means clustering on short time Fourier transform (STFT) features to detect healthy and malignant brain regions [201]. The texture and optical features of OCT images of human breast tissue at different resolutions were also investigated for the differentiation of major tissue types, such as adipose and malignant lesions. A relevance vector machine (RVM), a Bayesian framework of support vector machines, was used to perform classification of adipose tissue vs. solid tissue, and invasive ductal carcinoma (IDC) vs. normal stroma tissue with an overall accuracy of 84%, sensitivity of 89% and specificity of 71% [55]. Bareja et al., proposed a DL model to perform a binary classification of normal vs. cancerous breast tissue. The developed CNN achieved 96.74% accuracy, 92% sensitivity, and 99.73% specificity on the test dataset [202]. In addition, DL has been applied for breast tissue classification using a CNN for multi-class classification— i.e., adipose tissue, benign dense tissue, or malignant tissue—using multi-channel OCT images from 29 patients [203]. Various clinical studies investigated the feasibility of OCT in detecting basal cell carcinoma (BCC) of the skin without an automatic discrimination process [204,205]. Jorgensen et al. [39], attempted to automate the BCC detection process but with the use of features manually extracted from the OCT images, which limited the full automation of their method. Similarly, the approach of Schuh et al. [206] also required manual selection of regions of interest and thus was not fully automated. Recently, using a DL approach, a CNN was trained to discriminate BCC from normal skin with an accuracy of 95.93% (Figure 17) [207].

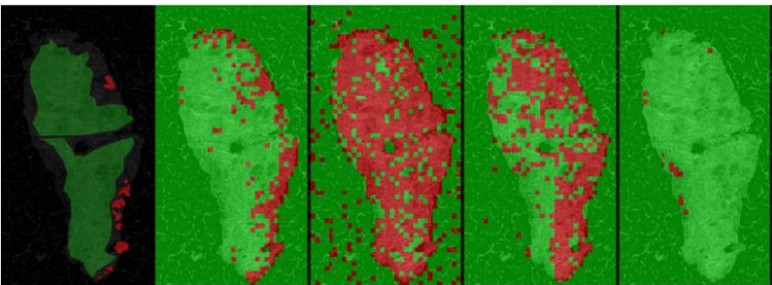

**Figure 17.** Ground truth labeling (left—black background) and network predictions comparison (from left to right): proposed method, InceptionV3 pre-trained, VGG16 pre-trained, VGG16 fine-tuned (green: normal, red: BCC). Reprinted with permission from [207]. Copyright 2018 IEEE: IEEE Proceedings.

## 6. Discussion

Oncology clinical trials, using OCT, increasingly highlight the promising potential of OCT for cancer identification and diagnosis, as well as monitoring therapy. However, OCT is still not as broadly utilized in large-scale oncology clinical applications as, for example, in ophthalmology. As in other areas with the potential of significant diagnostic and clinical impact, e.g., radiomics [208], this is due to limitations and technical challenges. These include inconsistencies in preprocessing and feature extraction, lack of strong feature correlation with biological biomarkers, small and imbalanced data sets, lack of open-access data, and the pitfalls of machine learning. For this reason, engineers and scientists are focusing on further developing OCT technology to enhance the clinical utility of OCT in early cancer diagnosis applications.

Detection of early malignancies will be significantly improved with the identification of cellular and sub-cellular features and increased contrast. Novel features such as GVD variations, which exist between normal and abnormal tissues, were shown to be a potential biomarker of disease. Novel techniques which measure the GVD in vivo have been proposed, have been demonstrated in the diagnoses of colon and esophageal cancer, and have the potential to provide diagnostically useful biomarkers [136,139]. The variations of the index of refraction, as well as the size of the cell nuclei, extracted from OCT data, have also been investigated for diagnostic purposes since they can reflect disease and cell dynamic changes [166,181]. However, large and detailed clinical studies are still required to prove the correlation of these features to both relevant biological changes but also to predictors of disease outcome.

Development of robust computational methods for feature extraction and selection and machine learning, which can effectively distinguish early abnormalities from normal tissues, are being actively developed and incorporated in the clinical workflow of OCT systems. Automated algorithms created for image segmentation, feature extraction and classification of in vivo OCT images of human tissues are becoming an integral part of OCT clinical studies. Traditional ML methods have provided promising results in differentiating benign vs. malignant or normal vs. cancerous lesions in various tissues. However, there are also some limitations in ML. Data undergo the same processing and so make the algorithms less robust. In addition, the pre-processing applied is not consistent between studies. Furthermore, human intervention, which is sometimes used, is risky as features are selected solely by the user. On the other hand, DL eliminates human intervention as feature extraction occurs internally to the networks. Additionally, the algorithms are more robust since data are not processed uniformly and the networks can work with large complex datasets from different modalities. However, there is still the risk of overfitting and uncertainty in the explainability of the network results which are not easy to interpret, hindering their clinical acceptance. A successful example of ML combined different features extracted from OCT images of human esophagus. Several ML algorithms were compared for their ability to discriminate various regions as normal, BE or dysplasia and resulted in 85% accuracy

(67% sensitivity and 95% specificity) in discriminating normal vs. abnormal esophageal tissue and 89%, accuracy (80% sensitivity and 91% specificity) in the classification of BE vs. dysplasia [182]. However, the small size of the datasets and the lack of open-access data make comparisons between studies as well as repeatability assessments very challenging.

## 7. Conclusions

The screening, early diagnosis and management of cancer remains a formidable clinical challenge. Therefore, there is a need for an inexpensive, accurate and minimally invasive screening method which highly correlates with biological biomarkers and can identify and predict the progression of pre-cancerous or dysplastic lesions. OCT has the potential to play such a role although it has not yet been widely adopted in clinical practice in oncology. In addition to technical hurdles, adoption of OCT is hindered by difficulties in image interpretation and by lack of demonstrable stratification and prediction capabilities. This review provides an overview of recent efforts to extract morphological and sub-resolution features from OCT images, which can serve as image-based biomarkers that correlate to biological predictors of disease. Such features, in combination with ML and DL approaches, could lead, in the future, to a tool that is sufficiently accurate and robust to enable the clinical acceptance of OCT as an effective modality in the management of cancer.

**Author Contributions:** Conceptualization, C.P. (Christos Photiou), M.K. and C.P. (Costas Pitris); formal analysis, C.P. (Christos Photiou); investigation, C.P. (Christos Photiou); resources, C.P. (Christos Photiou); writing—original draft preparation, C.P. (Christos Photiou) and C.P. (Costas Pitris); writing—review and editing, C.P. (Christos Photiou), M.K. and C.P. (Costas Pitris); visualization, C.P. (Christos Photiou) and C.P. (Costas Pitris); supervision, C.P. (Costas Pitris). All authors have read and agreed to the published version of the manuscript.

**Funding:** This research was funded by the European Union's Horizon 2020 research and innovation program under grant agreement No. 739551 (KIOS CoE) and from the Republic of Cyprus through the Directorate General for European Programs, Coordination and Development.

**Institutional Review Board Statement:** Not applicable.

**Informed Consent Statement:** Not applicable.

**Data Availability Statement:** Not applicable.

**Conflicts of Interest:** The authors declare no conflict of interest.

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
