# Peer review of "Extracting Morphological and Sub-Resolution Features from Optical Coherence Tomography Images, a Review with Applications in Cancer Diagnosis"

_photonics, doi:10.3390/photonics10010051_

Round 1

Reviewer 1 Report

Review of Photonics-2106089
Title: Extracting morphological and sub-resolution feature from optical coherence tomography images with applications in cancer diagnosis

Authors: C. Photiou et al.

This paper is a review paper to overview recent OCT studies aimed at investigating alterations in different cancerous organs obtained by several types of OCT technologies including FF-OCT and endoscopic OCT. The paper well introduces many OCT literatures for cancer diagnosis and delineates quantitative parameters used in the cancer studies involving sub-cellular change in tissue morphology, variations in nuclear size, refractive index. Moreover, the paper also presents new algorithms for tissue segmentation and feature extraction based on deep learning for probing the tumor in early stage. Taken together, this paper could be accepted with no major revision.

1. At line 17, sub-resolution is not proper. Change it to sub-cellular resolution.

2. In Figs. 1-3, I recommend for authors to re-plot them, especially, for interferograms and depth-profiles for TD, SD, SS-OCT, which are not corresponding to the probing beam through the tissue (This is right for 100% reflector as a sample). In Fig. 2, beam ray in the spectrometer is not proper.

3. In Fig. 2, ‘r’ of ‘Reference mirror’ is cut.

Author Response

  1. At line 17, sub-resolution is not proper. Change it to sub-cellular resolution.

        The change has been applied.

  1. In Figs. 1-3, I recommend for authors to re-plot them, especially, for interferograms and depth-profiles for TD, SD, SS-OCT, which are not corresponding to the probing beam through the tissue (This is right for 100% reflector as a sample). In Fig. 2, beam ray in the spectrometer is not proper.

         The figures were corrected to the reviewer’s specifications

  1. In Fig. 2, ‘r’ of ‘Reference mirror’ is cut.

         The error was corrected

Reviewer 2 Report

The authors submitted a well-written, comprehensive review concerning the extraction of morphological and sub-resolution features from Optical Coherence Tomography images for use in cancer diagnosis. This is a relevant topic that justifies a review paper, since several efforts were produced in the recent years in this research field. A comprehensive overview is, therefore, welcomed.

The paper is well-written, and the review is complete. My only comments focus on the discussion and conclusions sections. In my opinion, the authors could provide their views on the future translation of the reviewed research studies into the clinics, addressing key points of a roadmap for such translation. The need for better explainability of deep learning-based methods is mentioned but other topics, like the necessity of multicentric studies and longitudinal studies, or the importance of testing the techniques over different OCT systems could also be discussed. Although the reviewed techniques are still mainly in the research phase, translation into clinics is the desirable goal. When addressing translation, it can be valuable and timesaving to learn from the challenges that radiomics is currently facing. This type of discussion could be added, and, in my opinion, it would enhance the paper.

Author Response

The discussion was edited as requested.

Reviewer 3 Report

Congratulation to the authors, it was very interesting to read you paper. It is very well structured and gives a clear understanding of OCT applications in the clinical practice. I liked the fact that post-processing methods for enhancing OCT results were addressed, although it could have comprised even more details about chemometrics.

Overall, I think it is a good review that should be published in it's current form, with just a double-check of the grammar, as there are a few misspellings here and there:

- page 3, figure 2: check "mirro"

- page 4, line 113: correct "OCT imaging has a several"

- page 4, line 122: space needed before reference 7

- page 9, line 348: correct "Bacal"

- page 9, line 352: given that all other abbreviations are defined, even more than once in some cases, define ROI

- page 9, line 374: correct "lack of a stripe-like patterns"

- page 18, line 723: remove "it"

- page 21, line 834: insert "are"

Author Response

  1. page 3, figure 2: check "mirro"

         The typo was corrected

  1. page 4, line 113: correct "OCT imaging has a several"

         The error was corrected.

  1. page 4, line 122: space needed before reference 7

         The typo was corrected.

  1. page 9, line 348: correct "Bacal"

         The error was corrected.

  1. page 9, line 352: given that all other abbreviations are defined, even more than once in some cases, define ROI

         The definition was added

  1. page 9, line 374: correct "lack of a stripe-like patterns"

         The typo was corrected.

  1. page 18, line 723: remove "it"

         The word was removed.

  1. page 21, line 834: insert "are"

         The typo was corrected.
